# The CFTR Amplifier Nesolicaftor Rescues TGF-β1 Inhibition of Modulator-Corrected F508del CFTR Function

**DOI:** 10.3390/ijms231810956

**Published:** 2022-09-19

**Authors:** Charles Bengtson, Neerupma Silswal, Nathalie Baumlin, Makoto Yoshida, John Dennis, Sireesha Yerrathota, Michael Kim, Matthias Salathe

**Affiliations:** Department of Internal Medicine, University of Kansas Medical Center, Kansas City, KS 66160, USA

**Keywords:** cystic fibrosis, CFTR, ion channels, inflammation, airway epithelium

## Abstract

Highly effective cystic fibrosis transmembrane conductance regulator (CFTR) modulators have led to dramatic improvements in lung function in many people with cystic fibrosis (PwCF). However, the efficacy of CFTR modulators may be hindered by persistent airway inflammation. The cytokine transforming growth factor-beta1 (TGF-β1) is associated with worse pulmonary disease in PwCF and can diminish modulator efficacy. Thus, strategies to augment the CFTR response to modulators in an inflammatory environment are needed. Here, we tested whether the CFTR amplifier nesolicaftor (or PTI-428) could rescue the effects of TGF-β1 on CFTR function and ciliary beating in primary human CF bronchial epithelial (CFBE) cells. CFBE cells homozygous for F508del were treated with the combination of elexacaftor/tezacaftor/ivacaftor (ETI) and TGF-β1 in the presence and absence of nesolicaftor. Nesolicaftor augmented the F508del CFTR response to ETI and reversed TGF-β1-induced reductions in CFTR conductance by increasing the expression of *CFTR* mRNA. Nesolicaftor further rescued the reduced ciliary beating and increased expression of the cytokines IL-6 and IL-8 caused by TGF-β1. Finally, nesolicaftor augmented the F508del CFTR response to ETI in CFBE cells overexpressing *miR-145*, a negative regulator of CFTR expression. Thus, CFTR amplifiers, but only when used with highly effective modulators, may provide benefit in an inflamed environment.

## 1. Introduction

Cystic fibrosis (CF) is a life-limiting, autosomal recessive monogenic disorder affecting approximately 100,000 individuals worldwide. Pathogenic mutations in the *cystic fibrosis transmembrane conductance regulator* (*CFTR*) gene lead to aberrant chloride and bicarbonate secretion across epithelial tissues [1]. The most common CFTR mutation, F508del, leads to altered protein folding, poor intracellular trafficking, and eventual degradation in the endoplasmic reticulum [2]. At the airway surface, CFTR acts in concert with other apical ion channels, including epithelial sodium channels (ENaC), calcium-activated chloride channels (CaCC), and large conductance, Ca^2+^-activated, and voltage-gated K^+^ (BK) channels to regulate hydration of the airway surface liquid (ASL) to allow normal mucociliary clearance (MCC) [3,4]. Mutant CFTR effectuates defective MCC, causing a cycle of mucus stasis, hyperinflammation and microbial colonization leading to progressive lung function decline, which is the primary cause of morbidity and mortality in people with CF (PwCF). The cytokine transforming growth factor-beta 1 (TGF-β1) has been implicated in this hyperinflammatory process [5,6,7]. Activation of TGF-β1 signaling pathways leads to an inflammatory cascade and elevated levels of TGF-β1 have been associated with worse lung disease in PwCF [8,9]. Additionally, TGF-β1 is known to decrease *CFTR* mRNA levels by increasing the expression of *microRNA-145* (*miR-145*) [6,7,10], thereby diminishing the efficacy of CFTR modulators [10,11].

CFTR modulators are small molecules that correct the underlying defect in the mutant CFTR protein. In clinical trials and real-world studies, these therapies lead to improvements in lung function, exacerbation frequency and quality of life [12,13,14,15]. The triple combination of elexacaftor/tezacaftor/ivacaftor (ETI or Trikafta^®^) leads to significant correction of F508del CFTR function in vitro as well as remarkable improvements in lung function in many, but not all, PwCF carrying at least one copy of F508del [14]. We recently found that TGF-β1 greatly diminishes the efficacy of ETI in vitro and that poor sweat chloride concentration responses, a surrogate measure of CFTR function, correlated with elevated levels of TGF-β1 in the upper airways of PwCF upon initiation of ETI [11]. Although anti-inflammatory therapies can counter some of the effects of TGF-β1 in the CF airway epithelium in vitro [11,16], additional strategies may be required to improve the CFTR response to modulators in a TGF-β1 dominant inflammatory environment.

A relatively new class of modulators, termed amplifiers, seek to improve CFTR protein biosynthesis and thus augment the function of CFTR correctors and potentiators [17]. The amplifier nesolicaftor (also known as PTI-428) can increase *CFTR* mRNA stability and translation, independent of *CFTR* genotype. This is accomplished, at least in part, by the association of nesolicaftor with poly(rC)-binding protein 1 (PCBP1) [18]. PCBP1 binds to consensus sequences in the mRNA to promote stabilization and translation [19,20]. Previous studies have shown that nesolicaftor can further improve CFTR conductance afforded by earlier generation CFTR correctors and potentiators in clinically relevant in vitro model systems [21,22]. Nesolicaftor has also been investigated in clinical trials along with approved as well as novel CFTR modulators with purported beneficial, but only small, effects [23]. Here, we investigated whether nesolicaftor could improve the F508del CFTR response to ETI, a highly effective CFTR modulator, in the setting of the inflammatory mediator TGF-β1 in vitro. We show that nesolicaftor rescues the TGF-β1-mediated inhibition of ETI-corrected CFTR function in primary human F508del CF bronchial epithelial (CFBE) cells, likely through mRNA stabilization. Nesolicaftor further reverses TGF-β1-induced reductions in ciliary beating and increases in secreted cytokine levels indirectly through its effects on apical ion channel function. Thus, amplifiers may impart greater benefit under certain inflammatory conditions in the CF airway.

## 2. Results

### 2.1. Nesolicaftor Exerts Differential Effects on Modulator-Corrected CFTR Function in F508del CFTR 16HBEge Cells In Vitro

We investigated the effects of nesolicaftor on the modulator correction of F508del CFTR using the 16HBE gene edited (16HBEge) human bronchial epithelial cell line with the F508del CFTR mutation engineered by the Cystic Fibrosis Foundation Therapeutics Lab [24]. Confluent submerged monolayers of F508del CFTR 16HBEge cells were treated for 24 h with nesolicaftor (10 µM) and the combination of lumacaftor/ivacaftor (LI), tezacaftor/ivacaftor (TI), or elexacaftor/tezacaftor/ivacaftor (ETI) before short-circuit currents (I*_SC_*) were measured in Ussing chambers. Ivacaftor was added acutely in Ussing chambers to potentiate CFTR function in all experiments (Figure 1A).

Nesolicaftor significantly increased F508del CFTR function compared to controls (Figure 1B), consistent with a previous report [22]. Nesolicaftor caused a relative increase in F508del CFTR function by 70% in LI-treated cells compared to LI alone and by 29% in TI-treated cells compared to TI alone (Figure 1B). On the other hand, nesolicaftor failed to improve the F508del CFTR response to ETI (Figure 1B). However, ETI restored F508del CFTR function to near wild-type levels (Appendix A), suggesting that further modulation with the addition of nesolicaftor was not possible. Moreover, nesolicaftor did not significantly increase expression levels of *CFTR* mRNA in modulator-treated F508del CFTR 16HBEge cells compared to controls (Appendix A).

### 2.2. Nesolicaftor Augments ETI-Corrected F508del CFTR Function in Primary CFBE Cells In Vitro

We next sought to determine whether nesolicaftor exerts similar effects on ETI-corrected F508del CFTR function in primary human CF bronchial epithelial (CFBE) cells cultured at the air-liquid interface (ALI). Fully differentiated CFBE cells homozygous for F508del have minimal CFTR function that is significantly corrected by ETI (Appendix A). These cells were treated for 24 h with ETI in the presence or absence of nesolicaftor (10 µM). Nesolicaftor significantly increased ETI-corrected F508del CFTR function in primary CFBE cells (Figure 2A,B). Furthermore, CFBE cells treated with nesolicaftor showed a nearly threefold increase in expression levels of *CFTR* mRNA compared to cells treated with ETI alone (Figure 2C). The effects of nesolicaftor on ion channel function were not restricted to CFTR, as nesolicaftor further enhanced the conductance of both ENaC and CaCC, consistent with a previous report (Figure 2D,E) [22]. Interestingly, nesolicaftor increased CaCC function in CFBE cells even in the absence of ETI (Appendix A).

Nesolicaftor increased the mRNA expression of *sodium channel epithelial 1 subunit beta* (*SCNN1B*), but not *SCNN1A*, suggesting that nesolicaftor is not causing a global increase in ion channel expression (Figure 2F,G). As *SCNN1A* expression is more abundant than *SCNN1B*, *SCNN1B* is thought to be rate-limiting and increased expression of *SCNN1B* is sufficient to drive an increase in ENaC function [25]. Nesolicaftor further increased the mRNA expression of *anoctamin 1(ANO1)*, also known as *TMEM16A* (Figure 2H), an integral subunit of CaCC [26]. Overall, these data suggest that the ability of nesolicaftor to augment ETI-corrected F508del CFTR function differs between 16HBEge and primary CFBE cells. These data also provide further evidence that nesolicaftor is not specific for CFTR and that it regulates the expression of other ion channels to modulate their function in the airway.

### 2.3. Nesolicaftor Reverses TGF-β1-Induced Decreases in ETI-Corrected F508del CFTR Function and Ciliary Beating in Primary CFBE Cells In Vitro

The ability of nesolicaftor to increase *CFTR* mRNA expression in primary CFBE cells led us to investigate whether nesolicaftor could rescue the effects of TGF-β1 on CFTR function. Treatment of F508del CFTR CFBE cells with TGF-β1 (5 ng/mL) caused a significant reduction in ETI-corrected F508del CFTR function after 24 h (Figure 3A,B). On the other hand, TGF-β1 did not significantly affect ETI-corrected F508del CFTR conductance in 16HBEge F508del CFTR cells (Appendix A). TGF-β1 reduces CFTR function in part by reducing *CFTR* mRNA expression through the induction of *miR-145* [6,7,10]. However, TGF-β1 failed to reduce *CFTR* mRNA expression in 16HBEge cells (Appendix A), revealing differences in how *CFTR* expression is regulated in primary CFBE and 16HBEge cells.

Treatment of CFBE cells with nesolicaftor (10 µM) rescued the effects of TGF-β1 on ETI-corrected F508del CFTR function after 24 h (Figure 3B). Moreover, CFTR activity was significantly greater in ETI- and nesolicaftor-treated CFBE cells in the presence of TGF-β1 compared to ETI-treated CFBE cells in the absence of TGF-β1 (Figure 3B). Nesolicaftor further reversed the decrease in *CFTR* mRNA expression induced by TGF-β1 (Figure 3C). Interestingly, TGF-β1 caused a significant reduction in ENaC function, but did not affect CaCC activity (Figure 3D,E). However, nesolicaftor increased both ENaC and CaCC activities in the presence of TGF-β1 (Figure 3D,E). The net effect of ion channel dysfunction caused by TGF-β1 was a significant reduction in ciliary beating after 24 h (Figure 3F). However, nesolicaftor was able to rescue TGF-β1-induced decreases in ciliary beating in ETI-treated F508del CFTR CFBE cells (Figure 3F).

### 2.4. Nesolicaftor Reverses TGF-β1-Induced Inflammation in ETI-Treated Primary CFBE Cells In Vitro

Treatment of F508del CFTR CFBE cells with TGF-β1 (5 ng/mL) caused significant increases in the levels of secreted IL-6 and IL-8 proteins despite the presence of ETI (Figure 4). Nesolicaftor rescued the increase in IL-6 levels and partially reversed the increase in IL-8 levels induced by TGF-β1 after 24 h (Figure 4).

### 2.5. miR-145 Exposure Mimics TGF-β1 Effects, Which Are Reversed by Nesolicaftor

TGF-β1 reduces *CFTR* mRNA expression in CFBE cells via the induction of the microRNA *miR-145* [10]. Therefore, we tested whether nesolicaftor could reverse the effects of *miR-145* on CFTR function and mRNA expression. Primary homozygous F508del CFTR CFBE cells were infected with a *miR-145* lentiviral expression vector. Expression levels of *miR-145* were significantly elevated in *miR-145*-infected cells compared to controls (Appendix A). Nesolicaftor (10 µM) significantly increased CFTR function and expression levels of *CFTR* mRNA in ETI-treated CFBE cells overexpressing *miR-145* after 24 h (Figure 5A–C). ENaC and CaCC activities were further significantly increased by nesolicaftor in *miR-145*-infected cells (Figure 5D,E).

## 3. Discussion

Despite the marked impact of ETI on pulmonary disease in PwCF, the degree of response in lung function and sweat chloride is not uniform [14], and the long-term impact is still unclear. Extended studies of the modulator ivacaftor in those with the G551D mutation provide evidence that, after an initial increase, the lung function trajectory of those on ivacaftor approximates that of a non-modulator control group by year seven [27]. The underlying reasons for the diminishing efficacy of CFTR modulators are likely to be numerous. Airway inflammation influences the efficacy of modulators [10,11,28,29], yet the composition of the inflammatory milieu in the CF airway is likely to dictate whether it will have a beneficial or detrimental effect on modulator therapy. Although some inflammatory cytokines can augment the effects of ETI [28,29], the cytokine TGF-β1 negatively regulates CFTR expression and function, even in the presence of ETI. This occurs through TGF-β1-mediated increases in *miR-145* and *cyclooxygenase-2* expression [6,11]. This underscores the need to develop therapies that, when used in conjunction with CFTR modulators, can further restore CFTR function towards wild-type levels and maximize their beneficial effects under inflammatory conditions.

The first-in-class amplifier nesolicaftor increases *CFTR* mRNA stability in vitro and was further shown to increase CFTR protein expression in the nasal mucosa of study participants with CF [30]. Here, we showed that nesolicaftor can augment the F508del CFTR response to ETI in primary CFBE cells in vitro. Importantly, nesolicaftor was further able to rescue the inhibition of ETI-corrected F508del CFTR function caused by TGF-β1. Nesolicaftor increased the expression of *CFTR* mRNA, consistent with its function as an amplifier. Notably, nesolicaftor increased expression levels of *CFTR* mRNA by nearly five-fold in TGF-β1 exposed CFBE cells. TGF-β1 causes significant downregulation of *CFTR* mRNA expression in part by inducing the expression of *miR-145* [10]. However, it is unclear how nesolicaftor reverses the effects of TGF-β1, and thus *miR-145*, on *CFTR* mRNA expression and translation.

Previous studies found that nesolicaftor increases *CFTR* mRNA stability via its interactions with PCBP1 [18]. PCBP1 preferentially binds to CU-rich elements and can regulate alternative splicing, translation, and the RNA stability of many different genes [31]. Nesolicaftor binds directly to PCBP1, which binds to a PCBP1 consensus site located within an open reading frame (ORF) of *CFTR* mRNA to increase its stability [18]. Although this is likely to be a part of the mechanism by which nesolicaftor rescues the effects of TGF-β1 on *CFTR* mRNA expression, the 3′-UTR of *CFTR* contains many CU-rich regions that may act as PCBP1 binding sites. Additionally, the 3′-UTR of *CFTR* also contains many miRNA recognition elements (MREs), including one for *miR-145*. Thus, it is possible that a nesolicaftor-PCBP1 complex could sterically inhibit *miR-145* from binding as the suppression of the miRNA target is largely dependent on binding site accessibility. Although these experimental conditions were not tested, they warrant further investigation.

Given its interaction with PCBP1, it is unlikely that the action of nesolicaftor is specific for CFTR. Indeed, in addition to its effects on CFTR function, we also observed significant increases in ENaC and CaCC conductance, both in the presence and absence of TGF-β1. The effects of nesolicaftor are consistent with the recent report by Venturini et al. [22]. We show here that nesolicaftor increases mRNA expression levels of *SCNN1B*, but not *SCNN1A*, suggesting nesolicaftor is not exerting global effects on mRNA translation and/or stability. Nesolicaftor further increased the expression of *ANO1* mRNA, which is critical for CaCC function. Although increased CaCC function is expected to correlate with increased airway hydration and MCC [32], ENaC function is associated with dehydration, reduced ciliary beating (due to airway surface liquid volume loss) and MCC [33]. Nesolicaftor also reversed TGF-β-induced reduction in ENaC activity. However, the net effect was a rescue of ciliary beating, indicating restoration of airway surface liquid. Furthermore, nesolicaftor reversed the increase in secreted IL-6 (*p* < 0.05) and IL-8 (trend but non-significant) levels induced by TGF-β1. Future studies in this model system with physiologic outputs of MCC will better define the respective contributions of ion channels alternative to CFTR upon modulation by nesolicaftor.

Interestingly, although we saw significant, but possibly clinically unimportant, CFTR current improvements in the F508del CFTR 16HBEge cell line when exposed to nesolicaftor and earlier generation CFTR modulators, this effect was not seen with ETI. Additionally, there was no significant reduction in CFTR expression or function in response to TGF-β1 (Figure 1 and Appendix A). The mechanism underlying the differential response to nesolicaftor as well as TGF-β1 between the submerged 16HBEge cell line and primary CFBE cells, which are re-differentiated at the air-liquid interface, is unclear. Given that the level of wild-type CFTR current in 16HBE14o- cells was on par with ETI-corrected F508del CFTR current, it is likely that further increases with the addition of nesolicaftor were not possible [34]. Additionally, it is possible that the immortalization of the parent 16HBE14o- cell line and frequent passaging could contribute to both genetic and epigenetic changes [35]. As compared to primary cells, in immortal cell lines, several signaling pathways are modulated and are known to be phenotypically different [34]. In a proteomics study, mouse liver cell lines showed enhanced TGF-β1 signaling compared to primary hepatocytes, suggesting that cellular signaling pathways are modified to compete with the increased proliferation rate [34]. In our cell culture model, there is no change in *CFTR* mRNA expression in TGF-β1 treated 16HBEge cells, which could be due to altered signaling. This finding may have implications for the choice of in vitro model systems for future studies utilizing CFTR amplifiers.

In conclusion, our study is the first to show that CFTR amplifiers can improve the response to the modulator ETI, despite the presence of inflammation, in a relevant in vitro model system. This lends further support to the potential clinical benefits of CFTR amplifiers as an add-on therapy to current generation highly effective CFTR modulators. However, the non-specific effects of nesolicaftor potentially hinder the broad application of this therapy. Improving the specificity of amplifiers could potentially increase the clinical benefit of such a therapy, particularly in an inflamed airway environment. Moreover, the ability of a more specific amplifier would be beneficial for other CFTR mutations that are currently ineligible for modulator therapies.

## 4. Materials and Methods

### 4.1. Chemicals

Nesolicaftor (PTI-428; Cat. No. HY-111680) was acquired from MedChemExpress (Monmouth Junction, NJ, USA). Lumacaftor (VX-809; Cat. No. S1565), ivacaftor (VX-770; Cat. No. S1144), tezacaftor (VX-661; Cat. No. S7059), and elexacaftor (VX-445; Cat. No. S8851) were acquired from Selleck Chemicals LLC (Houston, TX, USA). Recombinant human TGF-β1 (Cat. No. 240-B) was acquired from R&D Systems (Minneapolis, MN, USA).

### 4.2. Lungs

CF bronchial epithelial (CFBE) cells were obtained from the explanted lungs of appropriately consented CF patients undergoing lung transplantation, approved by the Institutional Review Board (IRB) of the University of Miami.

### 4.3. Cell Culture

The F508del CFTR 16HBEge cell line was kindly provided by the Cystic Fibrosis Foundation Therapeutics Lab. 16HBEge cells were cultured on collagen coated dishes in Minimal Essential Medium (Cat. No. 11095072; ThermoFisher Scientific, Waltham, MA, USA) with 10% heat inactivated fetal bovine serum (Cat. No. 26400044; ThermoFisher Scientific, Waltham, MA, USA) and 1% penicillin-streptomycin (Cat. No. 15140-122; ThermoFisher Scientific, Waltham, MA, USA). After reaching 80–100% confluency, cells were plated on Collagen IV (Cat. No. C7521; Sigma-Aldrich, St. Louis, MO, USA) coated Transwell filters at a density of 2 × 10^5^ cells/cm^2^ and maintained at 37 °C and 5% CO_2_. 16HBEge cells were grown in liquid-liquid interface for one week and media was changed every other day. The wild-type 16HBE14o- cell line was acquired from the American Type Culture Collection (ATCC, Manassas, VA, USA).

ALI cultures of primary human CFBE cells were established as previously described [11,16,36,37]. Briefly, CFBE cells were expanded in PneumaCult^TM^-Ex Plus Medium (Cat. No. 05040; STEMCELL Technologies, Cambridge, MA, USA) supplemented with 50 µg/mL gentamycin (Cat. No. 15710-064; ThermoFisher Scientific, Waltham, MA, USA) and 2.5 µg/mL amphotericin B (Cat. No. 15290-026; ThermoFisher Scientific, Waltham, MA, USA). CFBE cells were expanded further in a Bronchial Epithelial Cell Growth Medium (BEGM) before being seeded on Transwell inserts (Cat. No. 3460; Corning, Corning, NY, USA) or Snapwell inserts (Cat. No. 3801; Corning, Corning, NY, USA) at a density of 2 × 10^5^ cells in ALI media. ALI media was prepared according to established protocols [38]. The cells were kept submerged in ALI media until confluency before exposing them to air. ALI media was replaced every other day and the apical surface was washed using DPBS (Cat. No. 21-030-CV; Corning, Corning, NY, USA). The cells were allowed to differentiate for four to six weeks before the experiments were carried out.

### 4.4. miR-145 Lentivirus Transduction

CFBE cells were seeded at a density of 5 × 10^4^ cells/well in BEGM on collagen coated Transwell inserts in a 24-well plate (Cat. No. 3470; Corning, Corning, NY, USA). On the day of transduction, the apical media was replaced with fresh media (100 µL) containing 8 µg/mL polybrene and 50 µL of human *miR-145* lentivirus (Cat. No. mir-LV116; Biosettia, San Diego, CA, USA) or infection control lentivirus (Cat. No. mir-LV000). After 24 h, lentivirus- and polybrene-containing media was replaced with fresh ALI media. The following day, transduced cells were selected with ALI media containing 1 µg/mL puromycin (Cat. No. A11138; ThermoFisher Scientific, Waltham, MA, USA) and maintained at a liquid-liquid interface for three to five days or until they reached confluency before exposure to air. ALI cultures of transduced cells were allowed to differentiate for three to four weeks in ALI media containing puromycin before experiments were performed.

### 4.5. Ussing Chambers

Ion channel activities from 16HBEge cells and fully differentiated CFBE cells were recorded in Ussing chambers as described previously [11]. Briefly, Snapwell or Transwell filters were mounted in Ussing chambers (EasyMount chamber; Physiologic Instruments, Reno, NV, USA) connected to a VCC MC8 voltage clamp unit (Physiologic Instruments, Reno, NV, USA). Solutions were maintained at 37 °C by heated water jackets and bubbled with CO_2_. CFTR-dependent short-circuit currents (I*_SC_*) were measured as the change in I*_SC_* after CFTR inhibition by 10 µM CFTR_inh_-172 (Cat. No. C2992; Sigma-Aldrich, St. Louis, MO, USA) in the presence of 10 µM amiloride (Cat. No. A7410; Sigma-Aldrich, St. Louis, MO, USA) following CFTR potentiation and stimulation by 1 µM ivacaftor (VX-770) and 10 µM forskolin (Cat. No. F3917; Sigma-Aldrich, St. Louis, MO, USA), respectively. CaCC was stimulated by 100 µM UTP (Cat. No. ab146222; Abcam, Cambridge, MA, USA).

### 4.6. Ciliary Beat Frequency (CBF)

The CBF of primary CFBE cells was recorded using a high-speed Basler acA645 camera (Basler, Ahrensburg, Germany) mounted on a Zeiss Axiovert running SAVA turnkey system (Ammons Engineering, Clio, MI, USA) as previously described [39]. CBF was analyzed using the individual region-of-interest (ROI) method by SAVA [40]. Ciliary beating was recorded 1–2 mm away from the center of the insert, free of influence by liquid meniscus, for 2 s at four different ROIs.

### 4.7. Quantitative PCR (qPCR)

Total RNA was isolated from 16HBEge and primary CFBE cells using the E.Z.N.A.^®^ Total RNA Kit (Omega Bio-tek, Norcross, GA, USA) or miRNeasy Mini Kit (Qiagen, Germantown, MD, USA). TaqMan™ MicroRNA Reverse Transcription Kit (ThermoFisher Scientific, Waltham, MA, USA) was used for *miR-145* cDNA synthesis. qPCR was performed using TaqMan Gene Expression Assays (ThermoFisher Scientific, Waltham, MA, USA) for *miR-145* (Assay ID 002278 assay name hsa-miR-145), *CFTR* (Hs00357011_m1), *ANO1* (Hs00216121_m1), *SCNN1A* (Hs01013032_m1), and *SCNN1B* (Hs01548617_m1). *U6* was used as a reference gene for *miR-145*. *GAPDH* was used as a reference gene for *CFTR*, *ANO1*, *SCNN1A*, and *SCNN1B*.

### 4.8. ELISA

Basolateral media samples were analyzed using the Ella Automated Immunoassay System and Simple Plex cartridge-based immunoassay for IL-6 and IL-8 (Bio-Techne Corp., Minneapolis, MN, USA).

### 4.9. Statistics

Data are presented as mean ± SEM and are considered statistically significant when *p* < 0.05. All data were analyzed for normality distribution using the Shapiro-Wilk test. Paired or unpaired *t*-tests were used to compare the two groups. The two groups were compared with Mann-Whitney (non-parametric) tests if data was not normally distributed. A one-way ANOVA was used to compare multiple groups followed by Holm-Sidak. A Kruskal-Wallis (non-parametric) test was used to compare multiple groups if data was not normally distributed.

## Figures and Tables

**Figure 1 ijms-23-10956-f001:**
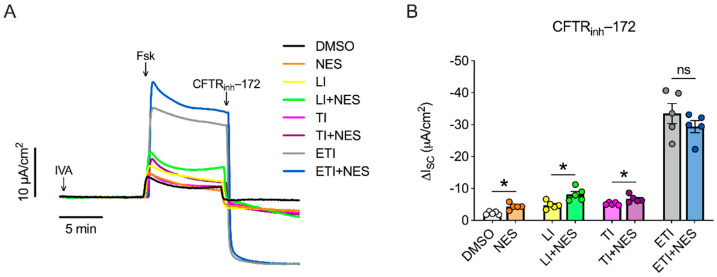
Effects of nesolicaftor (NES) on modulator-corrected F508del CFTR function in 16HBEge cells in vitro. (**A**) Confluent monolayers of F508del CFTR 16HBEge cells were treated with DMSO, NES (10 µM), lumacaftor (5 µM)/ivacaftor (1 µM) (LI), LI + NES, tezacaftor (5 µM)/ivacaftor (1 µM) (TI), TI + NES, elexacaftor (1 µM)/tezacaftor (5 µM)/ivacaftor (1 µM) (ETI), or ETI + NES both apically and basolaterally for 24 h. CFTR-dependent short-circuit currents (I*_SC_*) were measured in Ussing chambers as the change in I*_SC_* after CFTR_inh_-172 (10 µM) following forskolin (10 µM) stimulation in the presence of amiloride (10 µM) and ivacaftor (IVA; 1 µM). (**B**) Nesolicaftor alone increases F508del CFTR function. Nesolicaftor further increases F508del CFTR function in 16HBEge cells treated with LI or TI. Nesolicaftor does not improve F508del CFTR function in 16HBEge cells treated with ETI. n = 5. Statistics: * *p* < 0.05, Student’s *t*-test between treatment groups after assessing normality with Shapiro-Wilk. ns = not significant.

**Figure 2 ijms-23-10956-f002:**
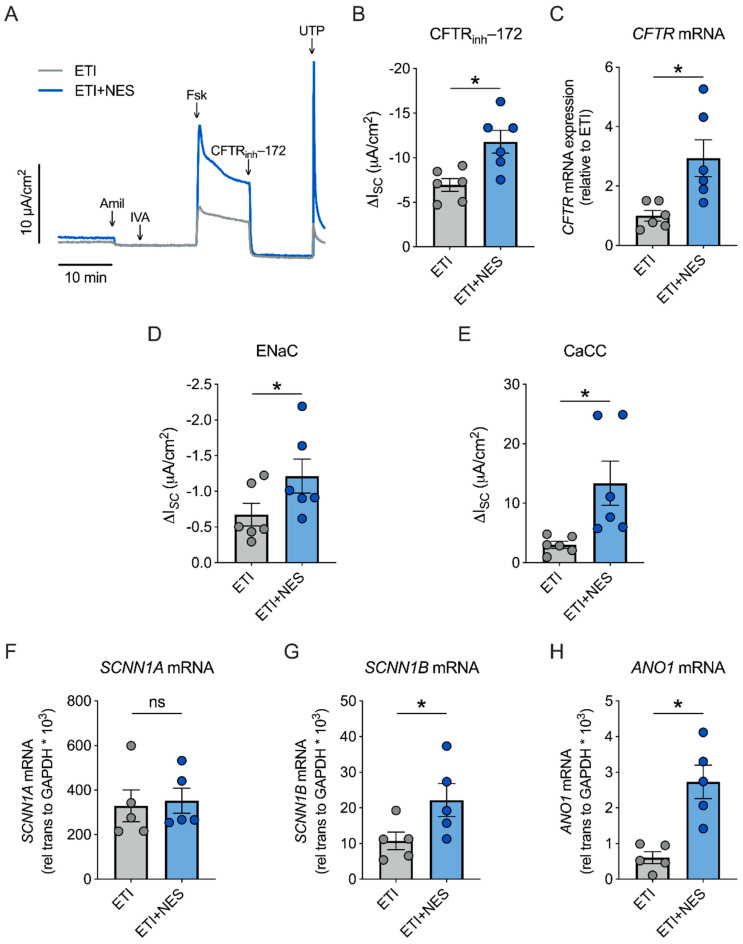
Nesolicaftor (NES) improves ETI-corrected F508del CFTR function in primary CFBE cells in vitro. (**A**) Fully differentiated ALI cultures of primary human CFBE cells homozygous for F508del were treated with elexacaftor (1 µM)/tezacaftor (5 µM)/ivacaftor (1 µM) (ETI) or ETI + NES (10 µM) for 24 h and CFTR-dependent I*_SC_* was measured in Ussing chambers. (**B**) Nesolicaftor significantly increases F508del CFTR function in ETI-treated CFBE cells after 24 h. n = 6, 3 CF lungs. (**C**) Nesolicaftor significantly increases expression levels of *CFTR* mRNA compared to ETI alone. n = 6, 3 CF lungs. (**D**,**E**) Amiloride-sensitive I*_SC_* (ENaC function) (**D**) and UTP-stimulated I*_SC_* (CaCC function) (**E**) are significantly increased by nesolicaftor in ETI-treated CFBE cells after 24 h. n = 6, 3 CF lungs. (**F**,**G**) Nesolicaftor significantly increases the expression of *SCNN1B*, but not *SCNN1A*, mRNA in ETI-treated CFBE cells after 24 h. n = 5, 3 CF lungs. (**H**) Nesolicaftor also causes a significant increase in the expression of *ANO1* mRNA in ETI-treated CFBE cells after 24 h. n = 5, 3 CF lungs. Statistics: * *p* < 0.05, Student’s *t*-test after assessing normality with Shapiro-Wilk.

**Figure 3 ijms-23-10956-f003:**
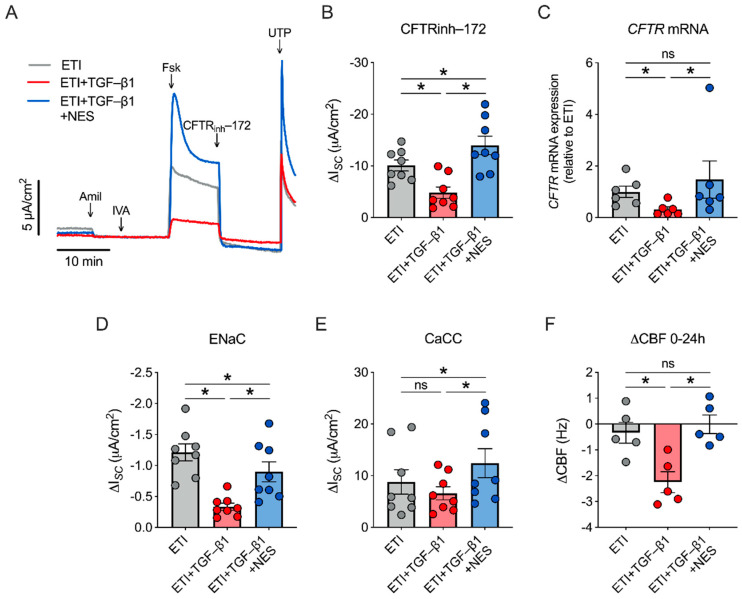
Nesolicaftor (NES) rescues TGF-β1-induced reductions in CFTR function and ciliary beating in primary CFBE cells in vitro. (**A**) Fully differentiated ALI cultures of primary human CFBE cells homozygous for F508del were treated with elexacaftor (1 µM)/tezacaftor (5 µM)/ivacaftor (1 µM) (ETI), ETI + TGF-β1 (5 ng/mL), or ETI + TGF-β1 + NES (10 µM) for 24 h and CFTR-dependent I*_SC_* was measured in Ussing chambers. (**B**) Nesolicaftor reverses the reduction in ETI-corrected F508del CFTR function caused by TGF-β1. n = 8, 3 CF lungs. (**C**) TGF-β1 significantly reduces levels of *CFTR* mRNA expression in ETI-treated CFBE cells. *CFTR* mRNA levels are restored by nesolicaftor. n = 6, 3 CF lungs. (**D**) Amiloride-sensitive I*_SC_* (ENaC function) is significantly reduced by TGF-β1. Nesolicaftor increases ENaC function in the presence of TGF-β1. n = 8, 3 CF lungs. (**E**) TGF-β1 does not significantly affect UTP-stimulated I*_SC_* (CaCC function). Nesolicaftor significantly increases CaCC function in the presence of TGF-β1. n = 8, 3 CF lungs. (**F**) TGF-β1 causes a significant reduction in ciliary beat frequency (CBF) in ETI-treated F508del CFTR CFBE cells after 24 h. TGF-β1-induced reductions in CBF are reversed by nesolicaftor. n = 5, 3 CF lungs. Statistics: * *p* < 0.05, one-way ANOVA followed by Holm-Sidak (**B**,**D**,**F**) or Friedman test (**C**,**E**) after assessing normality with Shapiro-Wilk. ns = not significant.

**Figure 4 ijms-23-10956-f004:**
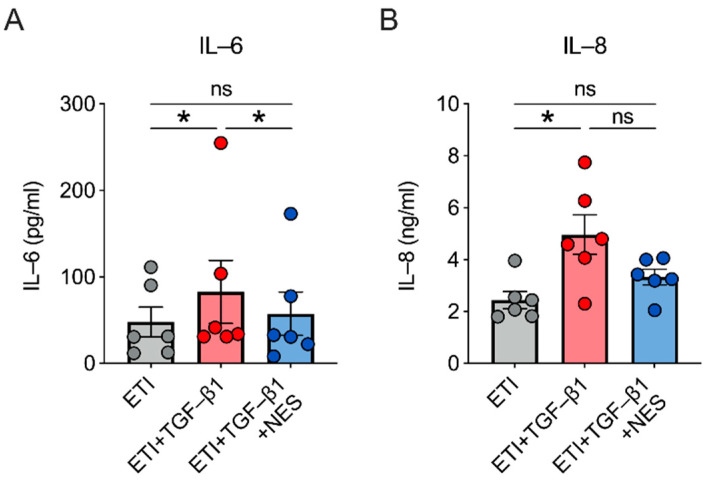
Nesolicaftor (NES) reverses TGF-β1-induced increases in IL-6 and IL-8 levels in primary CFBE cells in vitro. (**A**) Nesolicaftor rescues the increase in IL-6 protein levels induced by TGF-β1 in ETI-treated CFBE cells after 24 h. IL-6 expression was measured in the basolateral media. n = 6, 3 CF lungs. (**B**) Nesolicaftor led to a non-significant reduction in IL-8 protein levels induced by TGF-β1 in ETI-treated CFBE cells after 24 h. IL-8 levels were measured in the basolateral media. n = 6, 3 CF lungs. Statistics: * *p* < 0.05, Friedman test (**A**) or one-way ANOVA followed by Holm-Sidak (**B**) after assessing normality with Shapiro-Wilk. ns = not significant.

**Figure 5 ijms-23-10956-f005:**
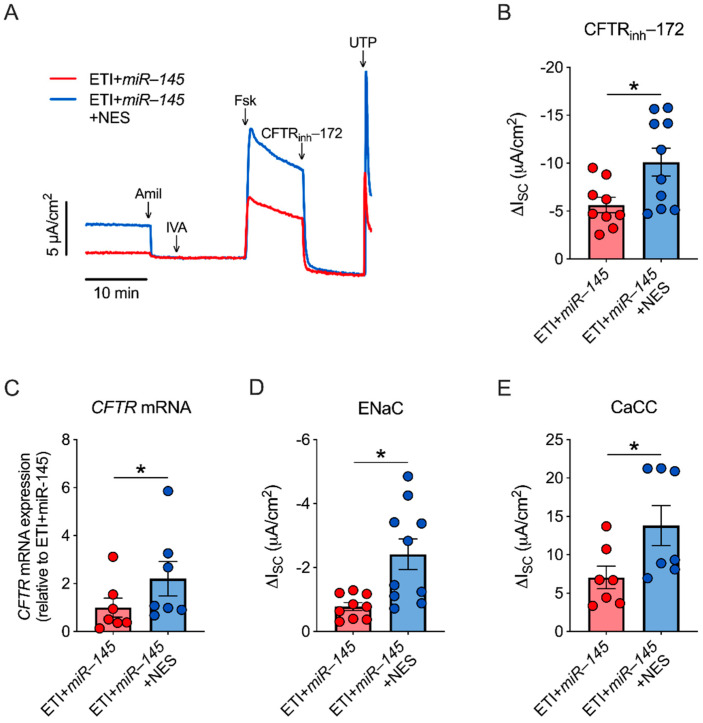
Nesolicaftor (NES) improves *miR-145*-induced reductions in CFTR function and mRNA expression in primary CFBE cells in vitro. (**A**) Primary human CFBE cells homozygous for F508del were infected with a *miR-145* lentiviral expression vector during differentiation. Fully differentiated ALI cultures overexpressing *miR-145* were treated with elexacaftor (1 µM)/tezacaftor (5 µM)/ivacaftor (1 µM) (ETI) or ETI + NES (10 µM) for 24 h and CFTR-dependent I*_SC_* was measured in Ussing chambers. (**B**) Nesolicaftor significantly increases ETI-corrected F508del CFTR function in *miR-145*-overexpressing CFBE cells after 24 h. n ≥ 9, 3 CF lungs. (**C**) Nesolicaftor significantly increases expression levels of *CFTR* mRNA compared to ETI + *miR-145* alone. n = 7, 3 CF lungs. (**D,E**) Amiloride-sensitive I*_SC_* (ENaC function) (**D**) and UTP-stimulated I*_SC_* (CaCC function) (**E**) are significantly increased by nesolicaftor in ETI-treated *miR-145*-overexpressing CFBE cells after 24 h. n ≥ 7, 3 CF lungs. Statistics: * *p* < 0.05, unpaired *t*-test (**B**,**D**), Wilcoxon test (**C**), or Mann-Whitney test (**E**) after assessing normality with Shapiro-Wilk.

## Data Availability

The data presented here are available upon request from the corresponding author.

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
