# Peer review of "The CFTR Amplifier Nesolicaftor Rescues TGF-β1 Inhibition of Modulator-Corrected F508del CFTR Function"

_ijms, 2022, doi:10.3390/ijms231810956_

Round 1

Reviewer 1 Report

I find this manuscript very interesting. The experiments were well performed and presented. There may be minor improvements possible. These are marked as comments by me directly in the manuscript.

Author Response

Response to reviewers

We appreciate the reviewers thoughtful and constructive comments regarding the initial submission of our manuscript. We have addressed, in a point-by-point manner, their comments below.

Reviewer 1:

  1. Inadequate discussion of nesolicaftor and its mechanism of action in the introduction

We have added a description of nesolicaftor’s currently known mechanism of action in the introduction to provide the reader a better understanding of proceeding sections of the manuscript.

  1. Mechanism of nesolicaftor rescuing TGF-b1 effects on ETI-corrected CFTR current and other physiologic outputs not mentioned in introduction.

We have added a description of mechanism in the discussion.

  1. Concern about different mechanisms of nesolicator’s effects in primary CFBE cells at the ALI and submerged 16HBEge treated with ETI.

We agree this difference was not well explained in the initial version of the manuscript. Given that ETI-corrected 16HBEge CFTR current is essentially the same as wild-type current, we surmise that maximal current has already been attained and the addition of nesolicaftor is unable to have further beneficial effects. This data has been added to the Supplementary Materials. We have also expanded discussion of possible differences between ALI cultures of primary CFBE cells and the 16HBEge cells lines in the response to ETI and TGF-b1 in the Discussion.

  1. Control condition missing from figures

We agree that control conditions should be addressed for all experiments. We have added supplemental figures showing control conditions for most experiments. However, matched DMSO and TGF-b1 only controls for comparisons of IL-6 and IL-8 expression with ETI and ETI+TGF-b1 are not available. However, in our previously published work (Kim MD et al., Am J Respir Crit Care Med, 2020), we found that TGF-b1 increased IL-8 levels as compared to controls in primary CFBE cells.

  1. Possibility that nesolicaftor negatively affected the stability of other drugs.

Although we did not directly test drug levels in these experiments, we feel this is unlikely. The dominant trend is that the addition of nesolicaftor increased the measured output as compared to ETI alone.

  1. Testing for ability of PCBP1-nesolicator to inhibit miR-145 binding

We agree that these experiments would provide a more mechanistic view of our results. However, these experiments are difficult to execute and beyond the scope of our manuscript. However, future studies evaluating this would be helpful to further understand this phenomenon.

Reviewer 2 Report

This paper examines the effect of the amplifier nesolicaftor on a set of CFTR cells, demonstrating a small effect (p<0.05, but overall probably better, as this was found for several effects) on F508del cells. This is perhaps clinically useful, so it is publishable in an issue devoted to CFTR. The work appears to be competently done, and the English is acceptable. The paper is a little difficult to read, given the number of acronyms and abbreviations that a reader not already working in the field must memorize to keep track of the results, but it is difficult to see how to avoid this.

This said, there is an intriguing finding buried in the paper, in Fig. 2D,E and some following work on EnaC and the CaCC channels. Why does nesolicaftor target the mRNA of channels, or does it—how general is the effect? One must assume that nesolicaftor cannot simply increase the output of all mRNA, increasing transcription of the entire genome. The authors, having put this result into the paper, have perhaps set forth a much more interesting puzzle.

In any case, this paper is acceptable, with perhaps minor improvements in presentation, so as to make it easier to follow. Obviously, this is a smaller step forward than most papers make, and it makes no attempt to advance understanding of the phenomenon it reports. However, this phenomenon is worth reporting, and it can be published.    

Author Response

Response to reviewers

We appreciate the reviewers thoughtful and constructive comments regarding the initial submission of our manuscript. We have addressed, in a point-by-point manner, their comments below.

Reviewer 2:

  1. Abbreviations hinder readability.

We agree that there is a large proportion of abbreviations in the manuscript. However, we also agree that this is unavoidable.

  1. Is the effect of nesolicaftor specific to the mRNA of channels or is this a more general effect?

Although PCBP1 is likely to target a large subset of genes, its effects on the expression of ion channel subunits is not entirely clear. We have added new data to show that nesolicaftor increases the mRNA expression of SCNN1B, but not SCNN1A, suggesting that nesolicaftor is not causing a global increase in the mRNA expression of channel subunits. Nesolicaftor also increases the mRNA expression of ANO1, which is critical for CaCC function. These data are included in a new Figure 2.